# Statistical tests under Dallal's model: Asymptotic and exact methods

Zhiming Li[1], Changxing Ma[2]*, Mingyao Ai[3]

**1** College of Mathematics and System Science, Xinjiang University, Urumqi, China, **2** Department of Biostatistics, University at Buffalo, Buffalo, NY, United States of America, **3** LMAM, School of Mathematical Sciences and Center for Statistical Science, Peking University, Beijing, China

* cxma@buffalo.edu

**Data Availability Statement:** All relevant data are within the manuscript.

**Funding:** This research is supported by the National Natural Science Foundation of China (Grant Nos: 12061070, 12071014, 11661076), and the Science and Technology Department of

## Abstract

This paper proposes asymptotic and exact methods for testing the equality of correlations for multiple bilateral data under Dallal's model. Three asymptotic test statistics are derived for large samples. Since they are not applicable to small data, several conditional and unconditional exact methods are proposed based on these three statistics. Numerical studies are conducted to compare all these methods with regard to type I error rates (TIEs) and powers. The results show that the asymptotic score test is the most robust, and two exact tests have satisfactory TIEs and powers. Some real examples are provided to illustrate the effectiveness of these tests.

## Introduction

In clinical medicine, we often encounter bilateral data taken from paired organs of patients such as eyes and ears. For the same patient, the intraclass correlation between responses of paired parts should be considered to avoid misleading results. There have been in the past various models to analyze such data. For example, Rosner [1] introduced a positive constant $R$ as a measure of the dependency by assuming that the probability of a response at one side of the paired body given a response at the other side is $R$ times to the response rate. Donner [2] provided an alternative approach and considered the common correlation coefficient in each of two groups. Under these two models, asymptotic and exact methods have been studied for many years and achieved significant progress.

Under Rosner's model, Tang et al. [3] developed exact and approximate procedures when sample size is small or the data structure is sparse. Qiu et al. [4] derived sample formulas for testing difference between two proportions. Shan and Ma [5], and Ma et al. [6] presented several asymptotic and exact methods to investigate the equality of proportions. Peng et al. [7] constructed asymptotic confidence intervals (CIs) of proportion ratio for correlated paired data. Under Donner's model, Pei et al. [8, 9] used asymptotic methods to analyze test statistics and CIs in two treated groups. Liu et al. [10, 11] provided exact methods to test the homogeneity of prevalence from multiple groups. Generally, asymptotic methods can produce empirical type I error rates (TIEs) close to the pre-specified nominal level for large samples. However,

Xinjiang Uygur Autonomous Region (Grant No: 2018Q011).

**Competing interests:** I have read the journal's policy and the authors of this manuscript have the following competing interests: The authors have declared that no competing interests exist.

they may yield inflation TIEs for small samples. Thus, exact tests become alternative to deal with the problem.

Dallal [12] indicated that Rosner's model may give a poor fit if the characteristic was almost certain to occur bilaterally with widely varying group-specific prevalence. Suppose the probability of response at one organ given response at the other organ was independent of its probability. He introduced likelihood ratio test for large samples. However, the approach performs poorly with unsatisfactory TIE control in small samples. Up to now, statistical inferences on Dallal's model have been less considered, including asymptotic and exact methods. This paper aims to propose asymptotic and exact methods for testing homogeneity of correlations among multiple bilateral data under Dallal's model.

The remainder of the work is organized as follows. In Section 2, we review bilateral data structure and introduce Dallal's model. The maximum likelihood estimations (MLEs) are derived for different hypotheses. Three asymptotic statistics and six exact procedures are proposed in Section 3. In Section 4, some numerical studies are conducted to compare these methods in terms of TIEs and power. Two examples are provided to illustrate these proposed approaches in Section 5. Conclusions are given in Section 6.

## Dallal's model and estimators

Suppose that $N$ patients is randomly allocated into $g$ groups. There are $m_i$ patients in the $i$th ($i = 1, \ldots, g$) group. Let $m_{li}$ be the number of patients who have $l(l = 0, 1, 2)$ organ(s) with improvement response(s) in the $i$th ($i = 1, \ldots, g$) group, and $S_l$ be the total number of patients with $l(l = 0, 1, 2)$ response(s). Obviously, $m_i = \sum_{l=0}^{2} m_{li}$ and $S_l = \sum_{i=1}^{g} m_{li}$. The data structure is shown in Table 1. Let $p_{li}$ be the probability that a patient has $l(l = 0, 1, 2)$ response(s) in the $i$th ($i = 1, \ldots, g$) group. The vector $\mathbf{m}_i \triangleq (m_{0i}, m_{1i}, m_{2i})^T$ follows a multinomial distribution $M(m_i; p_{0i}, p_{1i}, p_{2i})$. The probability density satisfies

$$f(\mathbf{m}_i) = \frac{m_i!}{m_{0i}! m_{1i}! m_{2i}!} p_{0i}^{m_{0i}} p_{1i}^{m_{1i}} p_{2i}^{m_{2i}}, \quad i = 1, \ldots, g.$$

Let $Z_{ijk} = 1$ if the $k$th organ of the $j$th patient has improvement response in the $i$th group for $k = 1, 2, i = 1, 2, \ldots, g$, and $j = 1, 2, \ldots, m_i$, and 0 otherwise. Under Dallal's model, we assume

$$P(Z_{ijk} = 1) = \pi_i, P(Z_{ijk} = 1 | Z_{ij(3-k)} = 1) = \gamma_i, \tag{1}$$

where $0 \leq \pi_i, \gamma_i \leq 1$. Especially, $\gamma_i = \pi_i$ means that two organ responses of each patient are completely independent, and $\gamma_i = 1$ represents that they are completely dependent in $i$th group. By using the Eq (1), the probabilities of no, one or both response(s) are

$$p_{0i} = 1 - (2 - \gamma_i)\pi_i, \quad p_{1i} = 2\pi_i(1 - \gamma_i), \quad p_{2i} = \pi_i\gamma_i, \quad i = 1, \ldots, g,$$

**Table 1. Bilateral data structure with $g$ groups.**

| Response ($l$) | Group | | | | | | Total |
|:---:|:---:|:---:|:---:|:---:|:---:|:---:|:---:|
| | 1 | 2 | $\ldots$ | $i$ | $\ldots$ | $g$ | |
| 0 | $m_{01}$ | $m_{02}$ | $\ldots$ | $m_{0i}$ | $\ldots$ | $m_{0g}$ | $S_0$ |
| 1 | $m_{11}$ | $m_{12}$ | $\ldots$ | $m_{1i}$ | $\ldots$ | $m_{1g}$ | $S_1$ |
| 2 | $m_{21}$ | $m_{22}$ | $\ldots$ | $m_{2i}$ | $\ldots$ | $m_{2g}$ | $S_2$ |
| Total | $m_1$ | $m_2$ | $\ldots$ | $m_i$ | $\ldots$ | $m_g$ | $N$ |

where $0 \leq p_{li} \leq 1$, $p_{01} + p_{1i} + p_{2i} = 1$, and $\max\left\{0, 1 - \frac{1}{2\pi_i}, 2 - \frac{1}{\pi_i}\right\} \leq \gamma_i \leq 1$. In the work, we are interested to test whether the correlations of $g$ groups are identical. Thus, the hypotheses are given by

$$H_0 : \gamma_1 = \cdots = \gamma_g = \gamma \text{ versus } H_1 : \gamma_i \neq \gamma_j \text{ for some } i \neq j \in \{1, 2, \ldots, g\}.$$

Denote $\mathbf{m} = (\mathbf{m}_1, \ldots, \mathbf{m}_g)$, $\boldsymbol{\pi} = (\pi_1, \ldots, \pi_g)$ and $\boldsymbol{\gamma} = (\gamma_1, \ldots, \gamma_g)$. Given the observation $\mathbf{m}$, the log-likelihood function

$$
\begin{aligned}
l(\boldsymbol{\pi}, \boldsymbol{\gamma}|\mathbf{m}) &= \ln\left(\prod_{i=1}^{g} f(\mathbf{m}_i)\right) \\
&= \sum_{i=1}^{g} [m_{0i} \ln(1 - 2\pi_i + \gamma_i \pi_i) + m_{1i} \ln(2\pi_i(1 - \gamma_i)) \\
&\quad + m_{2i} \ln(\pi_i \gamma_i)] + C,
\end{aligned}
\tag{2}
$$

where $C = \ln\left(\prod_{i=1}^{g} \frac{m_i!}{m_{0i}! m_{1i}! m_{2i}!}\right)$. Let $\hat{\pi}_i$ and $\hat{\gamma}_i$ be the unconstrained MLEs of $\pi_i$ and $\gamma_i$ under $H_1$. Differentiate (2) with respect to $\pi_i$ and $\gamma_i$, and set them to 0. The MLEs $\hat{\pi}_i$ and $\hat{\gamma}_i$ are the solutions of the following equations

$$
\begin{cases}
\dfrac{\partial l}{\partial \pi_i} = \dfrac{m_{0i}(\gamma_i - 2)}{1 - 2\pi_i + \pi_i \gamma_i} + \dfrac{m_{1i} + m_{2i}}{\pi_i} = 0, \\[3mm]
\dfrac{\partial l}{\partial \gamma_i} = \dfrac{m_{0i}\pi_i}{1 - 2\pi_i + \pi_i \gamma_i} - \dfrac{m_{1i}}{1 - \gamma_i} + \dfrac{m_{2i}}{\gamma_i} = 0.
\end{cases}
\tag{3}
$$

Then,

$$\hat{\pi}_i = \frac{m_{1i} + 2m_{2i}}{2m_i}, \quad \hat{\gamma}_i = \frac{2m_{2i}}{m_{1i} + 2m_{2i}}, \quad i = 1, \ldots, g. \tag{4}$$

Let $\tilde{\pi}_i$ and $\tilde{\gamma}$ be the constrained MLEs of $\pi_i$ and $\gamma$ under $H_0$. Similarly, they are the solution of the equations

$$
\begin{cases}
\dfrac{\partial l}{\partial \pi_i} = \dfrac{m_{0i}(\gamma - 2)}{1 - 2\pi_i + \pi_i \gamma} + \dfrac{m_{1i} + m_{2i}}{\pi_i} = 0, \\[3mm]
\dfrac{\partial l}{\partial \gamma} = \sum_{i=1}^{g}\left[\dfrac{m_{0i}\pi_i}{1 - 2\pi_i + \pi_i \gamma} - \dfrac{m_{1i}}{1 - \gamma} + \dfrac{m_{2i}}{\gamma}\right] = 0.
\end{cases}
$$

For the first equation, we have $\frac{m_{0i}}{1-2\pi_i+\pi_i\gamma} = \frac{m_{1i}+m_{2i}}{\pi_i(2-\gamma)}$. The second equation can be simplified as $\gamma (S_1 + 2S_2) - 2S_2 = 0$. Then, the constrained MLEs are obtained

$$\tilde{\pi}_i = \frac{(m_{1i} + m_{2i})(S_1 + 2S_2)}{2m_i(S_1 + S_2)}, i = 1, \ldots, g, \quad \tilde{\gamma} = \frac{2S_2}{S_1 + 2S_2}. \tag{5}$$

## Test methods

### An information matrix

Denote $\boldsymbol{\beta} = (\gamma_1, \ldots, \gamma_g, \pi_1, \ldots, \pi_g)$. According to the Eq (3), the second-order derivatives of $l$ with respect to $\pi_i$ and $\gamma_i$ are

$$\frac{\partial^2 l}{\partial \gamma_i^2} = -\frac{m_{2i}}{\gamma_i^2} - \frac{m_{1i}}{(\gamma_i - 1)^2} - \frac{m_{0i}\pi_i^2}{(\gamma_i\pi_i - 2\pi_i + 1)^2},$$

$$\frac{\partial^2 l}{\partial \gamma_i \partial \pi_i} = \frac{\partial^2 l}{\partial \pi_i \partial \gamma_i} = \frac{m_{0i}}{(\gamma_i\pi_i - 2\pi_i + 1)^2},$$

$$\frac{\partial^2 l}{\partial \pi_i^2} = -\frac{m_{1i} + m_{2i}}{\pi_i^2} - \frac{m_{0i}(\gamma_i - 2)^2}{(\gamma_i\pi_i - 2\pi_i + 1)^2}$$

for $i = 1, \ldots, g$, and $\frac{\partial^2 l}{\partial \gamma_i \partial \gamma_j} = \frac{\partial^2 l}{\partial \gamma_j \partial \gamma_i} = \frac{\partial^2 l}{\partial \gamma_i \partial \pi_j} = \frac{\partial^2 l}{\partial \pi_i \partial \gamma_j} = \frac{\partial^2 l}{\partial \pi_i \partial \pi_j} = \frac{\partial^2 l}{\partial \pi_j \partial \pi_i} = 0$ for $i \neq j$. Thus, the information matrix $\mathbf{I}_{\boldsymbol{\beta}}$ with respect to $\boldsymbol{\beta}$ is

$$\mathbf{I}_{\boldsymbol{\beta}} \triangleq \begin{bmatrix} \mathbf{I}_{11} & \cdots & 0 & \mathbf{I}_{(g+1)1} & \cdots & 0 \\ \vdots & \ddots & \vdots & \vdots & \ddots & \vdots \\ 0 & \cdots & \mathbf{I}_{gg} & 0 & \cdots & \mathbf{I}_{(2g)g} \\ \mathbf{I}_{1(g+1)} & \cdots & 0 & \mathbf{I}_{(g+1)(g+1)} & \cdots & 0 \\ \vdots & \ddots & \vdots & \vdots & \ddots & \vdots \\ 0 & \cdots & \mathbf{I}_{g(2g)} & 0 & \cdots & \mathbf{I}_{(2g)(2g)} \end{bmatrix},$$

where

$$\mathbf{I}_{ii} = -E\left(\frac{\partial^2 l}{\partial \gamma_i^2}\right) = -\frac{m_i\pi_i(\gamma_i - 2\pi_i + 1)}{\gamma_i(\gamma_i - 1)(\gamma_i\pi_i - 2\pi_i + 1)}, \quad i = 1, \ldots, g,$$

$$\mathbf{I}_{ij} = \mathbf{I}_{ji} = -E\left(\frac{\partial^2 l}{\partial \gamma_i \partial \pi_j}\right) = -\frac{m_i}{\gamma_i\pi_j - 2\pi_j + 1}, \quad (i,j) \in \{(1, g+1), \ldots, (g, 2g)\},$$

$$\mathbf{I}_{jj} = -E\left(\frac{\partial^2 l}{\partial \pi_j^2}\right) = -\frac{m_j(\gamma_j - 2)}{\pi_j(\gamma_j\pi_j - 2\pi_j + 1)}, \quad j = g+1, \ldots, 2g.$$

Otherwise, $I_{ij} = 0$. By calculation, its inverse matrix is

$$
\mathbf{I}_{\boldsymbol{\beta}}^{-1} = \begin{bmatrix} a_1 & \cdots & 0 & c_1 & \cdots & 0 \\ \vdots & \ddots & \vdots & \vdots & \ddots & \vdots \\ 0 & \cdots & a_g & 0 & \cdots & c_g \\ c_1 & \cdots & 0 & b_1 & \cdots & 0 \\ \vdots & \ddots & \vdots & \vdots & \ddots & \vdots \\ 0 & \cdots & c_g & 0 & \cdots & b_g \end{bmatrix}, \tag{6}
$$

where

$$
a_i = \frac{\gamma_i(\gamma_i - 1)(\gamma_i - 2)}{2m_i\pi_i}, \quad b_i = \frac{\pi_i(1 + \gamma_i - 2\pi_i)}{2m_i}, \quad c_i = -\frac{\gamma_i(\gamma_i - 1)}{2m_i}
$$

for $i = 1, \ldots, g$.

## Asymptotic test statistics

In this section, we propose three asymptotic tests for large samples based on the unconstrained and constrained MLEs.

(i) *Likelihood ratio test*. Let $\hat{\pi}_i, \hat{\gamma}_i$ be the unconstrained MLEs, and $\tilde{\pi}_i, \tilde{\gamma}$ be the constrained MLEs under $H_0$. Denote $\hat{\boldsymbol{\pi}} = (\hat{\pi}_1, \ldots, \hat{\pi}_g), \hat{\boldsymbol{\gamma}} = (\hat{\gamma}_1, \ldots, \hat{\gamma}_g)$ and $\tilde{\boldsymbol{\pi}} = (\tilde{\pi}_1, \ldots, \tilde{\pi}_g)$. Given observation $\mathbf{m}$, likelihood ratio statistic is given by

$$
T_L = 2\{l(\hat{\boldsymbol{\pi}}, \hat{\boldsymbol{\gamma}}|\mathbf{m}) - l_0(\tilde{\boldsymbol{\pi}}, \tilde{\gamma}|\mathbf{m})\},
$$

where $l(\boldsymbol{\pi}, \gamma|\mathbf{m})$ is defined in (2) and

$$
\begin{aligned}
l_0(\boldsymbol{\pi}, \gamma|\mathbf{m}) &= \sum_{i=1}^{g} [m_{0i}\ln(1 - 2\pi_i + \gamma\pi_i) + m_{1i}\ln(2\pi_i(1 - \gamma)) \\
&\quad + m_{2i}\ln\pi_i\gamma] + C.
\end{aligned}
$$

From (4) and (5), likelihood ratio test can be represented as

$$
T_L = 2\sum_{i=1}^{g}\ln\left[\left(\frac{m_{1i}}{S_1}\right)^{m_{1i}}\left(\frac{m_{2i}}{S_2}\right)^{m_{2i}}\left(\frac{S_1 + S_2}{m_{1i} + m_{2i}}\right)^{m_{1i} + m_{2i}}\right].
$$

(ii) *Score test*. Denote $U(\boldsymbol{\beta}) = \left(\frac{\partial l}{\partial\gamma_1}, \ldots, \frac{\partial l}{\partial\gamma_g}, 0, \ldots, 0\right)$. Under $H_0$, score test statistic can be defined as

$$
T_{SC} = U(\boldsymbol{\beta})I_{\boldsymbol{\beta}}^{-1}U(\boldsymbol{\beta})^{T}\Big|_{\gamma_1 = \gamma_2 = \cdots = \gamma_g = \tilde{\gamma}, \pi_i = \tilde{\pi}_i}.
$$

A direct calculation shows that the simplified form of $T_{SC}$ is

$$T_{SC} = \sum_{i=1}^{g} \frac{(S_1 m_{2i} - S_2 m_{1i})^2}{S_1 S_2 (m_{1i} + m_{2i})}.$$

(iii) *Wald-type test.* Let $\hat{\boldsymbol{\beta}} = (\hat{\gamma}_1, \ldots, \hat{\gamma}_g, \hat{\pi}_1, \ldots, \hat{\pi}_g)$. The null hypothesis $H_0: \gamma_1 = \ldots = \gamma_g$ is equivalent to $C\boldsymbol{\beta}^T = \mathbf{0}$, where $\mathbf{0}$ is a zero vector, and

$$C = \begin{pmatrix} 1 & -1 & 0 & \cdots & 0 & 0 & 0 & \ldots & 0 \\ 0 & 1 & -1 & \cdots & 0 & 0 & 0 & \ldots & 0 \\ \vdots & \vdots & \ddots & \ddots & \vdots & \vdots & \vdots & \ddots & \vdots \\ 0 & 0 & 0 & \cdots & 1 & -1 & 0 & \ldots & 0 \end{pmatrix}_{(g-1)\times(2g)}.$$

Hence, Wald-type test statistic can be written as

$$T_W = (\boldsymbol{\beta}C^T)(C\mathbf{I}_{\boldsymbol{\beta}}^{-1}C^T)^{-1}(C\boldsymbol{\beta}^T)|_{\boldsymbol{\beta}=\hat{\boldsymbol{\beta}}} = (\hat{\boldsymbol{\beta}}C^T)(C\mathbf{I}_{\hat{\boldsymbol{\beta}}}^{-1}C^T)^{-1}(C\hat{\boldsymbol{\beta}}^T),$$

where $\mathbf{I}_{\boldsymbol{\beta}}^{-1}$ is defined in (6). Let

$$\hat{a}_i = \frac{\hat{\gamma}_i(\hat{\gamma}_i - 1)(\hat{\gamma}_i - 2)}{2m_i\hat{\pi}_i} = \frac{4m_{1i}m_{2i}(m_{1i} + m_{2i})}{(m_{1i} + 2m_{2i})^4}, i = 1, \ldots, g. \tag{7}$$

Then,

$$CI_{\hat{\boldsymbol{\beta}}}^{-1}C^T = \begin{pmatrix} \hat{a}_1 + \hat{a}_2 & -\hat{a}_2 & 0 & \cdots & 0 & 0 \\ -\hat{a}_2 & \hat{a}_2 + \hat{a}_3 & -\hat{a}_3 & 0 & \cdots & 0 & 0 \\ 0 & -\hat{a}_3 & \hat{a}_3 + \hat{a}_4 & -\hat{a}_4 & \cdots & 0 & 0 \\ \vdots & \vdots & \ddots & \ddots & \ddots & \vdots & \vdots \\ 0 & 0 & \cdots & \cdots & \cdots & -\hat{a}_{g-1} & \hat{a}_{g-1} + \hat{a}_g \end{pmatrix}.$$

For convenience, denote $A = CI_{\hat{\boldsymbol{\beta}}}^{-1}C^T$. Obviously, $A$ is a symmetric tridiagonal matrix of order $g-1$. Let $d_{g-1} = \hat{a}_{g-1} + \hat{a}_g$, $d_j = \hat{a}_j + \hat{a}_{j+1} - \frac{\hat{a}_{j+1}^2}{d_{j+1}}$ for $j = 2, \ldots, g-1$, and $\delta_1 = \hat{a}_1 + \hat{a}_2$, $\delta_i = \hat{a}_i + \hat{a}_{i+1} - \frac{\hat{a}_i^2}{\delta_{i-1}}, i = g-2, \ldots, 1$. Following [13], $A^{-1}$ is also a symmetric matrix denoted by

$$A^{-1} = \begin{pmatrix} z_{11} & z_{12} & \cdots & z_{1(g-1)} \\ z_{12} & z_{22} & \cdots & z_{2(g-1)} \\ \vdots & \vdots & \ddots & \vdots \\ z_{1(g-1)} & z_{2(g-1)} & \cdots & z_{(g-1)(g-1)} \end{pmatrix},$$

where

$$z_{ij} = \begin{cases} \dfrac{(u_{i+1} \cdots u_j)(d_{j+1} \cdots d_j)}{\delta_i \cdots \delta_{g-1}}, & j > i, i,j = 1, \ldots, g-1, \\[2em] \dfrac{d_{i+1} \cdots d_{g-1}}{\delta_i \cdots \delta_{g-1}}, & i = 1, \ldots, g-1. \end{cases}$$

Since $\hat{\boldsymbol{\beta}} C^T = (\hat{\gamma}_1 - \hat{\gamma}_2, \hat{\gamma}_2 - \hat{\gamma}_3, \ldots, \hat{\gamma}_{g-1} - \hat{\gamma}_g)$, we obtain the simplified form

$$T_W = \sum_{i=1}^{g-1}\sum_{j=1}^{g-1}(\hat{\gamma}_i - \hat{\gamma}_{i+1})(\hat{\gamma}_j - \hat{\gamma}_{j+1})z_{ij}.$$

Next we provide the expressions of $T_W$ for $g = 2, 3, 4$. If $g = 2$, it follows that

$$T_W = \frac{(\hat{\gamma}_1 - \hat{\gamma}_2)^2}{\hat{a}_1 + \hat{a}_2} = \frac{4(m_{11}m_{22} - m_{12}m_{21})^2}{4(m_{11} + 2m_{21})^2(m_{12} + 2m_{22})^2 \sum_{i=1}^{2} \frac{m_{1i}m_{2i}(m_{1i}+m_{2i})}{(m_{1i}+2m_{2i})^4}}.$$

If $g = 3$, we have

$$T_W = \frac{\hat{a}_1(\hat{\gamma}_2 - \hat{\gamma}_3)^2 + \hat{a}_2(\hat{\gamma}_1 - \hat{\gamma}_3)^2 + \hat{a}_3(\hat{\gamma}_1 - \hat{\gamma}_2)^2}{\hat{a}_1\hat{a}_2 + \hat{a}_1\hat{a}_3 + \hat{a}_2\hat{a}_3}.$$

Denote $\hat{a} = \hat{a}_1\hat{a}_2\hat{a}_3 + \hat{a}_1\hat{a}_2\hat{a}_4 + \hat{a}_1\hat{a}_3\hat{a}_4 + \hat{a}_2\hat{a}_3\hat{a}_4$. If $g = 4$, then

$$\begin{aligned} T_W = \frac{1}{\hat{a}}[&(\hat{\gamma}_3 - \hat{\gamma}_4)((\hat{\gamma}_3 - \hat{\gamma}_4)(\hat{a}_1\hat{a}_2 + \hat{a}_1\hat{a}_3 + \hat{a}_2\hat{a}_3) + (\hat{\gamma}_2 - \hat{\gamma}_3)\hat{a}_3(\hat{a}_1 + \hat{a}_2) \\ &+ (\hat{\gamma}_1 - \hat{\gamma}_2)\hat{a}_2\hat{a}_3) + (\hat{\gamma}_1 - \hat{\gamma}_2)((\hat{\gamma}_1 - \hat{\gamma}_4)\hat{a}_2\hat{a}_3 + (\hat{\gamma}_1 - \hat{\gamma}_3)\hat{a}_2\hat{a}_4 \\ &+ (\hat{\gamma}_1 - \hat{\gamma}_2)\hat{a}_3\hat{a}_4) + (\hat{\gamma}_2 - \hat{\gamma}_3)((\hat{\gamma}_2 - \hat{\gamma}_4)\hat{a}_1\hat{a}_3 + (\hat{\gamma}_1 - \hat{\gamma}_4)\hat{a}_2\hat{a}_3 \\ &+ (\hat{\gamma}_2 - \hat{\gamma}_3)\hat{a}_1\hat{a}_4 + (\hat{\gamma}_1 - \hat{\gamma}_3)\hat{a}_2\hat{a}_4)], \end{aligned}$$

where $a_i$ is defined in (7).

Under $H_0$, test statistic $T_l(= T_L, T_{SC}$ or $T_W)$ has asymptotic chi-square distribution with $g - 1$ degrees of freedom. Let $\chi^2_{g-1,1-\alpha}$ be the $(1 - \alpha)$th quantile of the chi-square distribution with $g - 1$ degree of freedom. Given the nominal level $\alpha$, the null hypothesis $H_0$ will be rejected if the value of $T_l$ is larger than $\chi^2_{g-1,1-\alpha}$.

## Exact methods

Given the observed data $\mathbf{m} = (\mathbf{m}_1, \ldots, \mathbf{m}_g)$, let $T_l(\mathbf{m})$ be the value of the aforementioned statistic $T_l(l = L, SC, W)$. The asymptotic (A) $p$-values of these statistics are defined by

$$p_l^A(\mathbf{m}^*) = P(T_l(\mathbf{m}^*) \le \chi^2_{g-1,1-\alpha}), \quad l = L, SC, W, \tag{8}$$

where $\mathbf{m}^*$ is an observed data of $\mathbf{m}$. For convenience, we call $p_L^A, p_{SC}^A$ and $p_W^A$ as "A approach" based on statistics $T_L, T_{SC}$ and $T_W$. Asymptotic tests work well when the sample size is large. However, they have some limitations if the sample size is relatively small. Several exact conditional and unconditional methods are proposed for small samples based on these statistics.

An exact conditional method is introduced under the assumption that all of $m_i(i = 1, \ldots, g)$ and $S_l(l = 0, 1, 2)$ are fixed in Table 1. Thus, the cell values of the table follow a hypergeometric

distribution. Define the tail area of statistics $T_L$, $T_{SC}$ and $T_W$ as

$$\Psi_l(\mathbf{m}^*) = \{\mathbf{m} : T_l(\mathbf{m}) \geq T_l(\mathbf{m}^*), \mathbf{m} \in S(\mathbf{m}^*)\}, \quad l = L, SC, W,$$

where $S(\mathbf{m}^*) = \{\mathbf{m} : S_l = S_l^*, l = 0, 1, 2\}$. According to the tail area $\Psi_l(\mathbf{m}^*)$, the exact conditional (C) $p$-values can be calculated by

$$p_l^C(\mathbf{m}^*) = \sum_{\mathbf{m} \in \Psi_l(\mathbf{m}^*)} \left( \frac{\prod_{i=1}^{g} \frac{m_i!}{m_{0i}!m_{1i}!m_{2i}!}}{\frac{N!}{S_0^*!S_1^*!S_2^*!}} \right), \quad l = L, SC, W. \tag{9}$$

Here, $p_L^C, p_{SC}^C$ and $p_W^C$ are described as "C approach" based on statistics $T_L$, $T_{SC}$ and $T_W$.

Another exact $p$-value is from Basu's maximization approach [14]. It can be obtained by maximizing the tail probability over all nuisance parameters instead of the constrained MLEs under $H_0$. In this case, we define the tail area of statistic $T_l(l = L, SC, W)$ as

$$\Omega_l(\mathbf{m}^*) = \{\mathbf{m} : T_l(\mathbf{m}) \geq T_l(\mathbf{m}^*)\}, \quad l = L, SC, W$$

for a given table $\mathbf{m}^*$. Denote $\Theta = \{\boldsymbol{\pi}: \pi_i \in [0, 1], i = 1, \ldots, g\}$ and

$$\Lambda = \left\{ \boldsymbol{\gamma} : \max\left\{0, 1 - \frac{1}{2\pi_i}, 2 - \frac{1}{\pi_i}\right\} \leq \gamma_i \leq 1, \pi_i \in [0, 1], i = 1, \ldots, g \right\},$$

where $\boldsymbol{\pi} = (\pi_1, \ldots, \pi_g)$ and $\boldsymbol{\gamma} = (\gamma_1, \ldots, \gamma_g)$. Hence, under maximization (M) method, three exact unconditional $p$-value of are given by

$$p_l^M(\mathbf{m}^*) = \sup_{\boldsymbol{\pi} \in \Theta, \boldsymbol{\gamma} \in \Lambda} \left\{ \sum_{\mathbf{m} \in \Omega_l(\mathbf{m}^*)} L(\boldsymbol{\pi}, \boldsymbol{\gamma}|\mathbf{m}) \right\}, \quad l = L, SC, W, \tag{10}$$

where $L(\boldsymbol{\pi}, \boldsymbol{\gamma}|\mathbf{m}) = \exp(l(\boldsymbol{\pi}, \boldsymbol{\gamma}|\mathbf{m}))$ and $l(\boldsymbol{\pi}, \boldsymbol{\gamma}|\mathbf{m})$ is defined in (2). Corresponding to "A approach" and "C approach", $p_L^M, p_{SC}^M$ and $p_W^M$ are called "M approach" based on $T_L$, $T_{SC}$ and $T_W$.

## Numerical studies

In this section, we investigate the performance of the proposed asymptotic and exact tests in terms of TIEs and powers under different parameter settings.

We first compare asymptotic methods $T_L$, $T_{SC}$ and $T_W$ with empirical TIEs. Let $g = 2, 3, 4, \pi = 0.3: 0.02: 0.5, \gamma = 0.3: 0.02: 0.8$ and $m = m_1 = \cdots = m_g = 15, 50, 100$. Here, $a: b: c$ means increasing from $a$ to $c$ by $b$. For each parameter setting, 10,000 samples are randomly generated from the null hypothesis $H_0$. Given the nominal level $\alpha = 0.05$, empirical TIE is calculated by the proportion of rejecting $H_0$, i.e., the number of rejections/10,000. Figs 1, 2 and 3 show the distribution surfaces of empirical TIEs for all the tests under $\pi_i = \pi$ and $\gamma_i = \gamma (i = 1, 2, \ldots, g; g = 2, 3, 4)$. According to Tang et al. [3], a test is *liberal* if its empirical TIE is greater than 0.06, *conservative* if it is less than 0.04, otherwise it is *robust*. We observe that score test is more robust than other tests since its TIEs are closer to the pre-determined level $\alpha = 0.05$. All the tests work well for larger sample size. However, likelihood ratio and Wald-type tests have inflated TIEs and are especially liberal when sample size is small. Some of their TIEs may be less than 0.04 or greater than 0.06.

Next we calculate the empirical powers of these tests according to the parameter settings for $m = 15, 50, 100$: (i) $g = 2, \boldsymbol{\pi} = (0.2, 0.3), \gamma_1 = 0.2: 0.05: 0.95, \gamma_2 = 0.1$, (ii) $g = 3, \boldsymbol{\pi} = (0.2, 0.3, 0.3), \gamma_1 = 0.2: 0.05: 0.95, \gamma_2 = \gamma_3 = 0.1$, and (iii) $g = 4, \boldsymbol{\pi} = (0.2, 0.3, 0.3, 0.3), \gamma_1 = 0.2: 0.05: 0.95, \gamma_2 = \gamma_3 = \gamma_4 = 0.1$. For each parameter setting, we randomly choose 10,000 samples from the

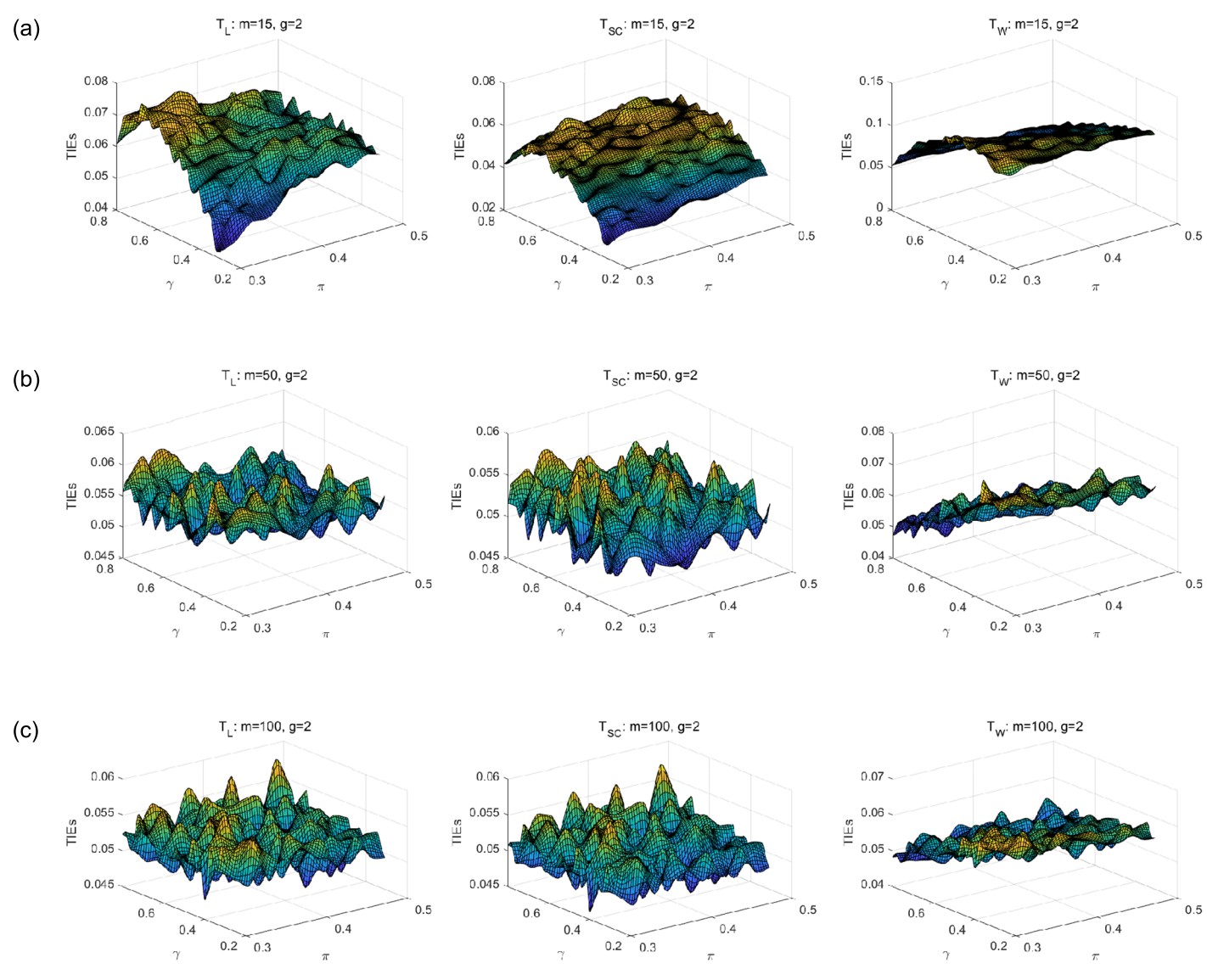

**Fig 1. Empirical TIE surfaces of asymptotic tests for $g = 2$, $\pi_i = \pi$ and $\gamma_i = \gamma$.**

alternative hypothesis $H_1$. The empirical power is computed by the proportion of rejecting $H_0$ for all samples. Fig 4 reflects the empirical powers of three proposed tests for $g = 2, 3, 4$. The powers will increase when sample size is larger or the group number increases. Especially, the powers of all the tests are very close when $m = 50, 100$. However, there exists some differences between these tests for smaller samples. Wald-type test has higher power and likelihood ratio test has lower power.

Considering the limitations of asymptotic methods, we analyse $A$, $C$ and $M$ approaches for small samples. Unlike 10,000 random samples of asymptotic tests, we need to generate all possible tables with random cell values. For $m = 10$ and $g = 2, 3$, there are totally 4,356 and 287,492 tables. The TIEs and powers are obtained for $m = m_1 = \cdots = m_g = 10$ and $g = 2, 3$ according to the cases: $\pi = 0$: 0.04: 1, $\gamma = 0$: 0.04: 1, satisfying $0 \leq p_{li} \leq 1 (l = 0, 1, 2, i = 2, 3)$. At the given nominal level $\alpha = 0.05$, the probabilities are calculated by the log-likelihood (2) of all possible tables. We will reject the null hypothesis $H_0$ if the probability is less than 0.05. Figs 5 and 6

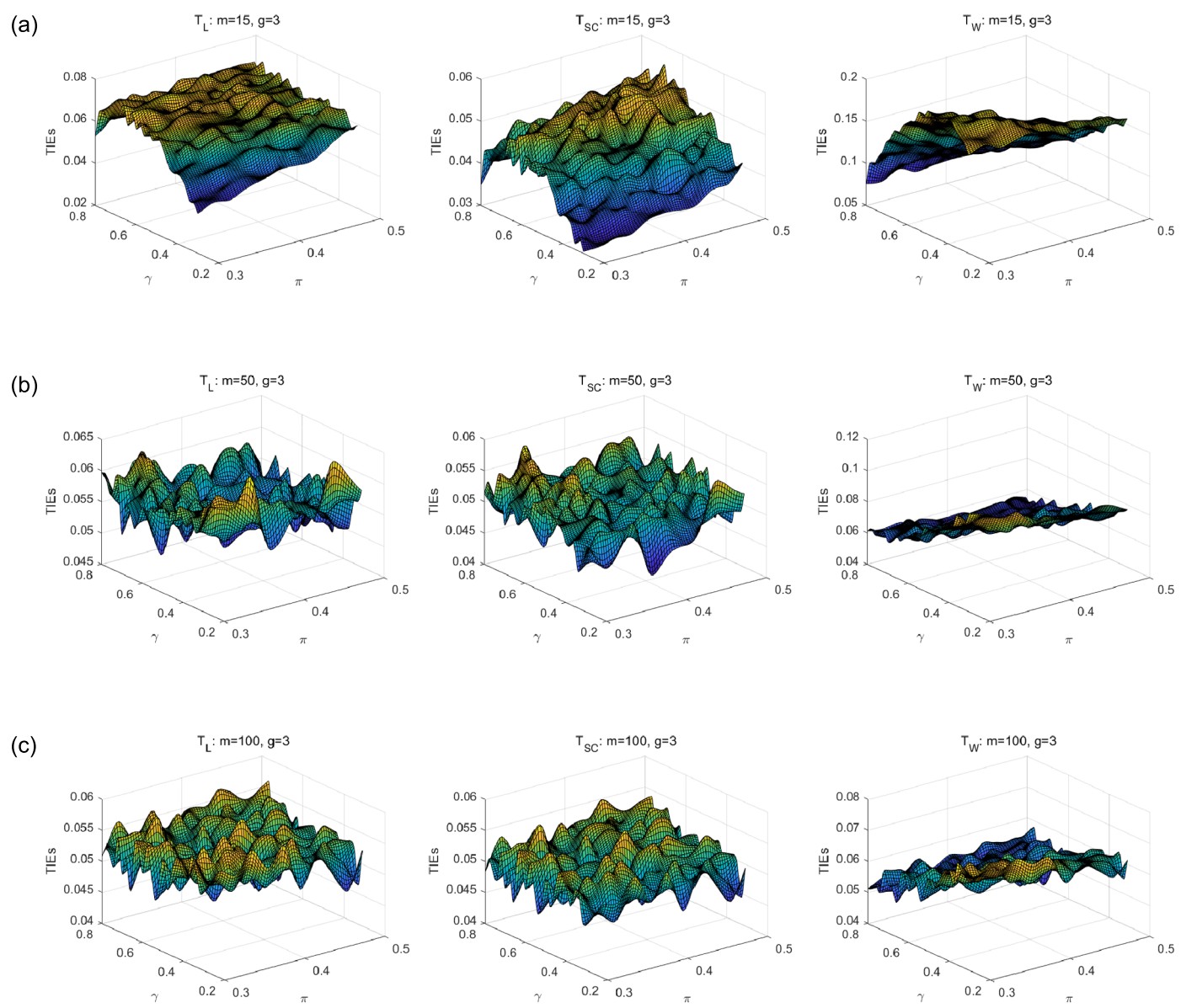

**Fig 2. Empirical TIE surfaces of asymptotic tests for $g = 3$, $\pi_i = \pi$ and $\gamma_i = \gamma$.**

show TIE surfaces of all the exact methods for $\pi_i = \pi$ and $\gamma_i = \gamma$ ($i = 1, \ldots, g; g = 2, 3$). We observe that $A$ approach $p_{SC}^A$ is closer to the pre-specified nominal level $\alpha = 0.05$ for $m = 10$ and $g = 2, 3$. However, $p_L^A$ and $p_W^A$ have the inflated TIEs. For $C$ approach, $p_W^C$ is better than $p_L^C$ and $p_{SC}^C$ since they have the inflated TIEs. The $M$ approaches $p_L^M$ and $p_{SC}^M$ can produce satisfactory TIEs.

Fig 7 provides the powers of exact methods according to parameter settings for $m = 10$: (i) $g = 2$, $\boldsymbol{\pi} = (0.2, 0.3)$, $\gamma_1 = 0.2$: 0.05: 0.9 and $\gamma_2 = 0.1$, and (ii) $g = 3$, $\boldsymbol{\pi} = (0.2, 0.3, 0.3)$, $\gamma_1 = 0.2$: 0.05: 0.9 and $\gamma_2 = \gamma_3 = 0.1$. We observe that the powers will increase when $m$ or $\gamma_1$ increases under other fixed parameters. The powers of $A$, $C$ and $M$ approaches are relatively close based on statistics $T_l (l = L, SC)$.

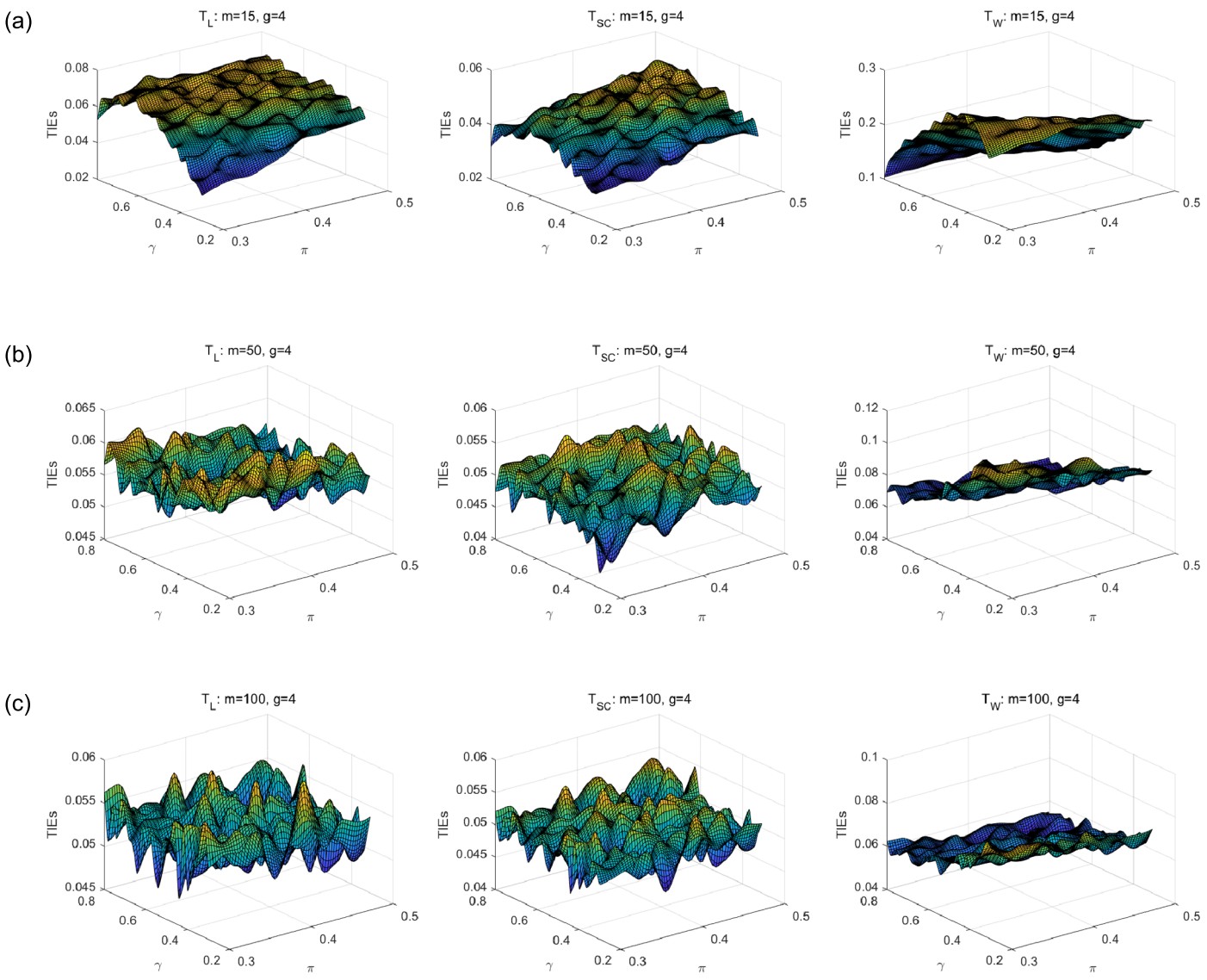

**Fig 3. Empirical TIE surfaces of asymptotic tests for $g = 4$, $\pi_i = \pi$ and $\gamma_i = \gamma$.**

Note that all parameter settings of asymptotic and exact methods are studied under balanced designs, that is, $m = m_1 = \cdots = m_g$. For unbalanced case, we can handle it through some examples.

## Real examples

In this section, two real examples with unbalanced designs are provided to illustrate our proposed methods at the nominal level $\alpha = 0.05$. We first show an example with large samples based on asymptotic test statistics.

**Example 1** [15] There were 216 patients aged 20-39 with retinitis pigmentosa (RP) at the Massachusetts Eye and Ear infirmary. They were divided into four genetic groups (Table 2): autosomal dominant RP (DOM), autosomal recessive RP (AR), sex-linked RP (SL) and isolate RP (ISO).

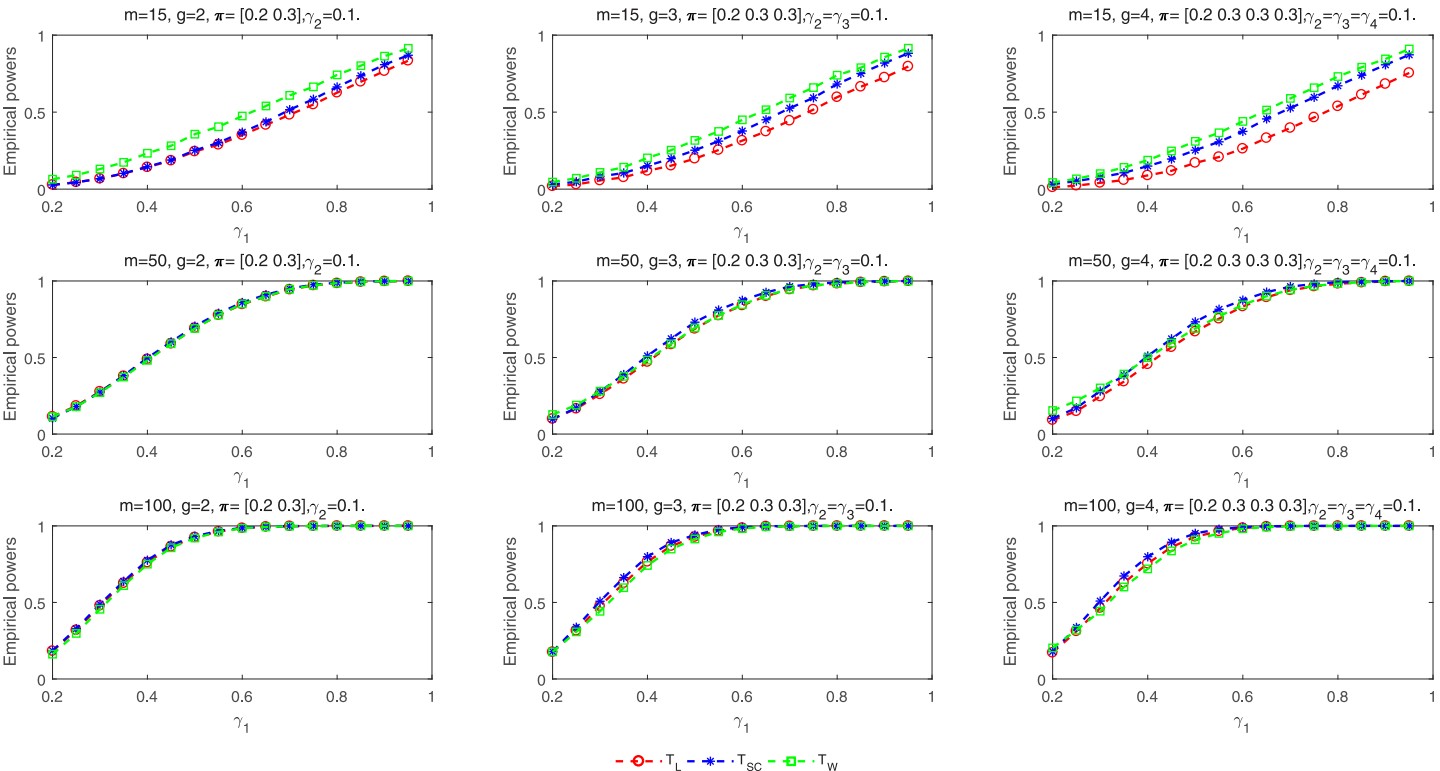

**Fig 4. Empirical power curves of asymptotic tests for $g$ = 2, 3, 4.**

Let $m_{li}$ be the number of patients with $l(l = 0, 1, 2)$ affected eyes in the $i$th ($i = 1, 2, 3, 4$) group. Under Dallal's model, we are interested to test if the correlations of these four groups are equal, i.e., $H_0 : \gamma_1 = \gamma_2 = \gamma_3 = \gamma_4 \triangleq \gamma$. Table 3 provides the results of statistics, $p$-values and constrained MLEs. Moreover, the unconstrained MLEs $\hat{\boldsymbol{\pi}} = (\hat{\pi}_1, \hat{\pi}_2, \hat{\pi}_3, \hat{\pi}_4) = (0.3571, 0.5476, 0.7895, 0.4662)$ and $\hat{\boldsymbol{\gamma}} = (\hat{\gamma}_1, \hat{\gamma}_2, \hat{\gamma}_3, \hat{\gamma}_4) = (0.7000, 0.7826, 0.9333, 0.8261)$. Given the nominal level $\alpha = 0.05$, $T_L$, $T_W$, $T_{SC} < \chi^2_{3,0.95} = 7.81$ and $p$-values are greater than 0.05. Thus, there is no evidence to reject $H_0$. That is to say, the correlations of four groups are equal: $\gamma_1 = \gamma_2 = \gamma_3 = \gamma_4 = 0.8246$.

For small sample case, we provide another example to compare the effectiveness of asymptotic and exact methods.

**Example 2** [16] A double-blind clinical trial was conducted to study amoxicillin treatment of acute otitis media with effusion (OME) in twenty-four children at 14 days. Each child underwent no, unilateral or bilateral OME and was assigned into three groups according to ages: <2, 2-5 and $\leq$6 years (Table 4). Denote $\mathbf{m}^* = (2, 2, 11, 5, 1, 3, 6, 0, 7)$. Next we apply asymptotic and exact methods to test $H_0$: $\gamma_1 = \gamma_2 = \gamma_3 = \gamma$.

Through calculating (4) and (5), the unconstrained MLEs $\hat{\boldsymbol{\pi}} = (0.8000, 0.3889, 0.1179)$, $\hat{\boldsymbol{\gamma}} = (0.9167, 0.8571, 0.9524)$, and the constrained MLEs $\tilde{\boldsymbol{\pi}} = (0.7924, 0.4064, 0.1417)$, $\tilde{\gamma} = 0.9063$ under $H_0$. Then, $T_L(\mathbf{m}^*) = 0.2445$, $T_{SC}(\mathbf{m}^*) = 0.2525$ and $T_W(\mathbf{m}^*) = 0.2285$. Table 5 provides the comparison of asymptotic and exact methods. The result shows that there is no significant difference among the correlations of two groups regardless of any approaches.

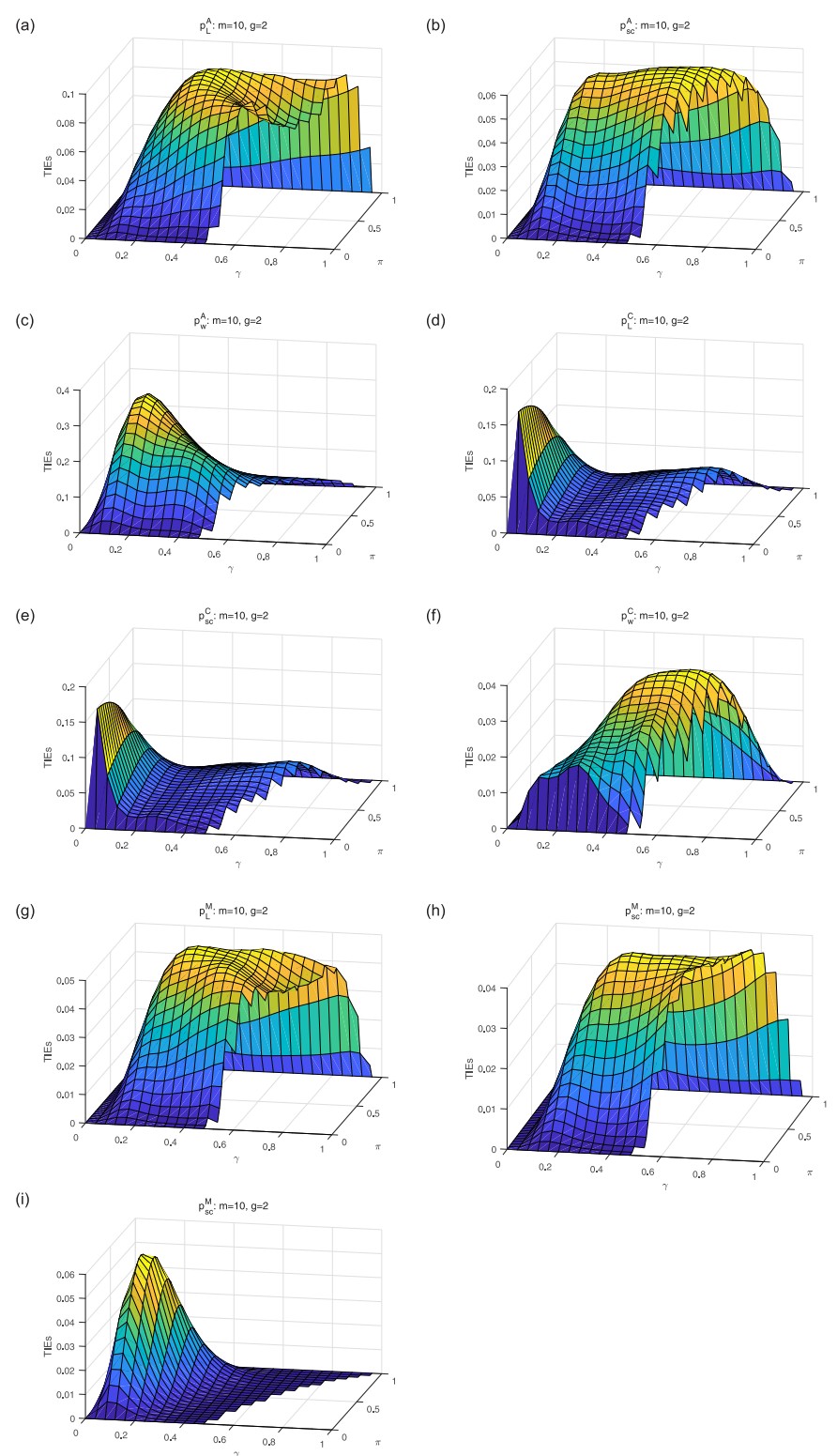

**Fig 5. TIE surfaces of exact approaches for $m = 10$, $g = 2$, $\pi_i = \pi$ and $\gamma_i = \gamma$.**

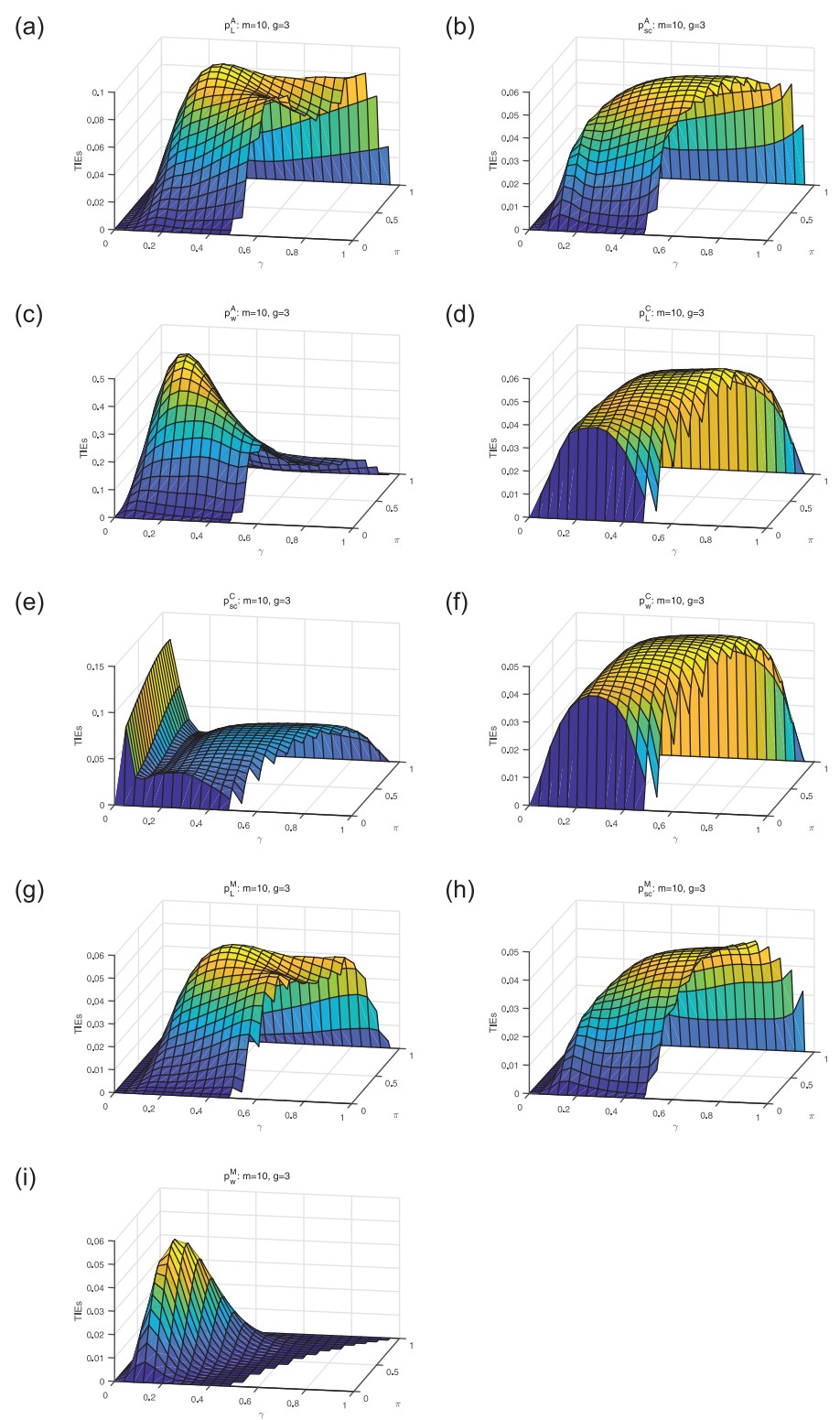

**Fig 6. TIE surfaces of exact approaches for $m = 10$, $g = 3$, $\pi_i = \pi$ and $\gamma_i = \gamma$.**

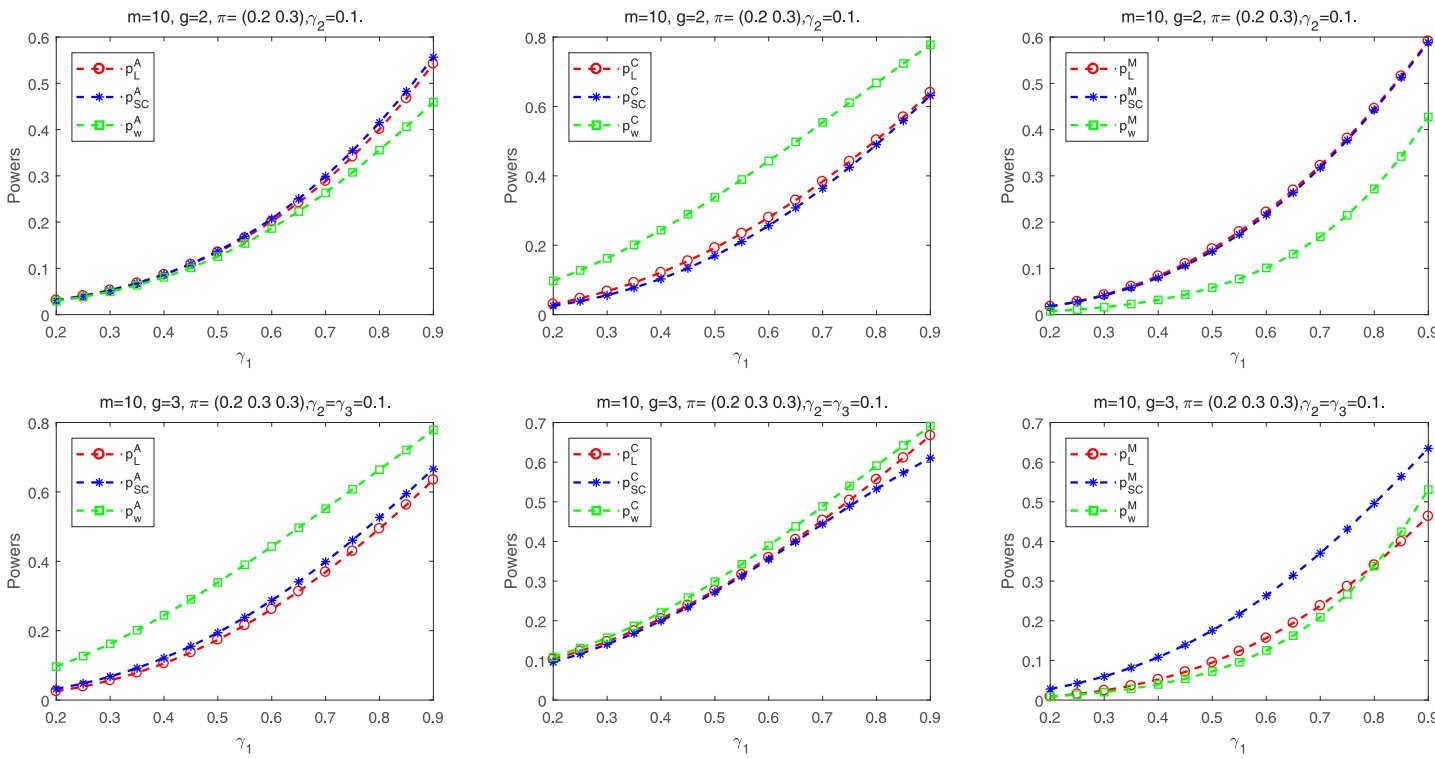

**Fig 7. Power curves of exact approaches for $m = 10$ and $g = 2, 3$.**

**Table 2. The number of patients for genetic types.**

| Response | DOM | AR | SL | ISO | Total |
|---|---|---|---|---|---|
| 0 | 15 | 7 | 3 | 67 | 92 |
| 1 | 6 | 5 | 2 | 24 | 37 |
| 2 | 7 | 9 | 14 | 57 | 87 |
| Total | 28 | 21 | 19 | 148 | 216 |

**Table 3. Test statistics, $p$-values and constrained MLEs under $H_0$.**

| Value | Test statistics | | | $\tilde{\pi} = (\tilde{\pi}_1, \tilde{\pi}_2, \tilde{\pi}_3, \tilde{\pi}_4)$ | $\tilde{\gamma}$ |
|---|---|---|---|---|---|
| | $T_L$ | $T_{SC}$ | $T_W$ | | |
| Statistic value | 4.4569 | 4.1831 | 5.7594 | (0.3950, 0.5675, 0.7165, 0.4656) | 0.8246 |
| $p$-value | 0.2162 | 0.2424 | 0.1239 | | |

**Table 4. 14-day OME status.**

| Response | < 2 years | 2-5 years | ≥6 years | Total |
|---|---|---|---|---|
| 0 | 2 | 5 | 6 | 13 |
| 1 | 2 | 1 | 0 | 3 |
| 2 | 11 | 3 | 1 | 15 |
| Total | 15 | 9 | 7 | 31 |

**Table 5. Comparison of asymptotic and exact *p*-values.**

| Method | A approach | | | C approach | | | M approach | | |
|---|---|---|---|---|---|---|---|---|---|
| | $p_L^A$ | $p_{SC}^A$ | $p_W^A$ | $p_L^C$ | $p_{SC}^C$ | $p_W^C$ | $p_L^M$ | $p_{SC}^M$ | $p_W^M$ |
| *p*-value | 0.8849 | 0.8814 | 0.8920 | 0.8643 | 0.9022 | 0.7686 | 0.8963 | 0.9899 | 0.8042 |

## Conclusions

In this paper, we propose asymptotic statistics and exact procedures to test if the correlations of multiple bilateral data are equal under Dallal's model. Three asymptotic test statistics are likelihood ratio $T_L$, score $T_{SC}$ and Wald-type $T_W$ for large sample. The explicit expressions of these tests are obtained, and their asymptotic *p*-values $p_l^A (l = L, SC, W)$ are denoted by $A$ approach. For small sample, six exact methods are derived based on statistics $T_L$, $T_{SC}$ and $T_W$, including three conditional exact $C$ procedures $p_l^C (l = L, SC, W)$ and three unconditional exact $M$ approaches $p_l^M (l = L, SC, W)$.

Numerical studies are conducted to investigate the performance of asymptotic and exact methods in terms of TIEs and powers. When the samples is larger, empirical TIEs and powers of $T_L$, $T_{SC}$ and $T_W$ are close to each other. In general, score test $T_{SC}$ is more robust than other two tests. However, these tests may produce unacceptable TIEs such as Wald-type test when the samples is smaller. The results are similar to those of Rosner's and Donner's models, see Ma et al. [6] and Liu et al. [10]. For small sample, we obtain TIE surfaces and power curves of exact $C$ and $M$ approaches with two and three groups, comparing with $A$ approach. As for TIEs, the $A$ approaches $p_L^A$ and $p_W^A$ are liberal, and $p_{SC}^A$ is close to the nominal level 0.05 under different parameter configurations. The $C$ approaches $p_L^C$ and $p_{SC}^C$ tend to be more inflated than $p_W^C$. The $M$ approach $P_{SC}^M$ is better than $p_L^M$ and $p_W^M$. On the other hand, the powers of exact methods are very close based on likelihood ratio $T_L$ and score $T_{SC}$. For $C$ approach, $p_W^C$ has higher power, while $p_l^C (l = L, SC)$ has lower power. Moreover, $p_{SC}^M$ has higher power, but $p_W^M$ has lower power in $M$ approach.

The ideas of asymptotic and exact methods can be extended other data structures with larger or small samples such as crash data. For example, Zeng et al. [17–19] proposed some models for the analysis of crash rates by injury severity. Dong et al. [20] introduced mixed logit model to investigate the difference between single- and multi-vehicle accident probability. Chen et al. [21–23] analyzed unbalance panel models by using real-time environmental and traffic big data. For these problems, we will leave these for future research.

## Acknowledgments

The authors thanks the editor and referees for constructive comments that help improve the manuscript.

## Author Contributions

**Funding acquisition:** Zhiming Li.

**Methodology:** Zhiming Li.

**Software:** Zhiming Li.

**Writing – original draft:** Zhiming Li, Changxing Ma.

**Writing – review & editing:** Changxing Ma, Mingyao Ai.

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
