## [Decision Letter · Decision Letter 0]

28 Sep 2020

PONE-D-20-26283

Statistical tests under Dallal's model: Asymptotic and exact methods

PLOS ONE

Dear Dr. Li,

Thank you for submitting your manuscript to PLOS ONE. After careful consideration, we feel that it has merit but does not fully meet PLOS ONE’s publication criteria as it currently stands. Therefore, we invite you to submit a revised version of the manuscript that addresses the points raised during the review process.

We look forward to receiving your revised manuscript.

Kind regards,

Feng Chen

Academic Editor

PLOS ONE

Journal Requirements:

2. In your Data Availability statement, it is unclear why you have selected 'No - some restrictions will apply' option. PLOS defines a study's minimal data set as the underlying data used to reach the conclusions drawn in the manuscript and any additional data required to replicate the reported study findings in their entirety. All PLOS journals require that the minimal data set be made fully available. For more information about our data policy, please see http://journals.plos.org/plosone/s/data-availability.

3. Please note that PLOS ONE does not allow for the use of footnotes in its publications. As such, we ask you to remove all footnotes and move the information contained in them to the main text.

4. Please update your submission to use the PLOS LaTeX template. The template and more information on our requirements for LaTeX submissions can be found at http://journals.plos.org/plosone/s/latex.

Reviewers' comments:

Reviewer's Responses to Questions

**Comments to the Author**

1. Is the manuscript technically sound, and do the data support the conclusions?

Reviewer #1: Yes

Reviewer #2: Yes

2. Has the statistical analysis been performed appropriately and rigorously? 

Reviewer #1: Yes

Reviewer #2: Yes

3. Have the authors made all data underlying the findings in their manuscript fully available?

Reviewer #1: No

Reviewer #2: Yes

4. Is the manuscript presented in an intelligible fashion and written in standard English?

Reviewer #1: Yes

Reviewer #2: Yes

5. Review Comments to the Author

Reviewer #1: This study proposes some asymptotic and exact methods for testing the equality of correlations for multiple bilateral data under Dallal's model. Their performance is compared via two numerical studies. The paper is generally well organized and written. A minor suggestion is that more references on the MLE should be acknowledged, such as:

A multivariate random parameters Tobit model for analyzing highway crash rate by injury severity. Accident Analysis and Prevention, 2017, 99: 184-191.

Jointly modeling area-level crash rates by severity: A Bayesian multivariate random-parameters spatio-temporal Tobit regression. Transportmetrica A: Transport Science, 2019, 15(2): 1867-1884.

Spatial joint analysis for zonal daytime and nighttime crash frequencies using a Bayesian bivariate conditional autoregressive model. Journal of Transportation Safety and Security, 2020, 12(4): 566-585.

Besides, some more directions for future research are suggested to draw in the Conclusion Section.

Reviewer #2: The topic of this paper is interesting. The methods sound. The results are meaningful and useful. There is one suggestion to improve this paper.

Some related references about likelihood ratio test or maximum likelihood estimations could be added.

[1] Investigating the Differences of Single- and Multi-vehicle Accident Probability Using Mixed Logit Model, Journal of Advanced Transportation, 2018, UNSP 2702360.

[2] Analysis of hourly crash likelihood using unbalanced panel data mixed logit model and real-time driving environmental big data. 2018, JOURNAL OF SAFETY RESEARCH. 65: 153-159.

[3] Investigation on the Injury Severity of Drivers in Rear-End Collisions Between Cars Using a Random Parameters Bivariate Ordered Probit Model, International Journal of Environmental Research and Public Health, 2019, 16(14) , 2632.

[4] Crash Frequency Modeling Using Real-Time Environmental and Traffic Data and Unbalanced Panel Data Models, International Journal of Environmental Research and Public Health, 2016, 13(6), 609.

6. PLOS authors have the option to publish the peer review history of their article (what does this mean?). If published, this will include your full peer review and any attached files.

Reviewer #1: No

Reviewer #2: No

---

## [Author Response · Author response to Decision Letter 0]

2 Nov 2020

Reviewer 1:

This study proposes some asymptotic and exact methods for testing the equality of correlations for multiple bilateral data under Dallal's model. Their performance is compared via two numerical studies. The paper is generally well organized and written. A minor suggestion is that more references on the MLE should be acknowledged, such as:

[1] Zeng Qiang, Wen Huiying, Huang Helai, Pei Xin, Wong S.C. A multivariate random parameters Tobit model for analyzing highway crash rate by injury severity. Accident Analysis and Prevention, 2017, 99: 184-191. https://doi.org/10.1016/j.aap.2016.11.018.

[2] Qiang Zeng, Qiang Guo, S. C. Wong, Huiying Wen, Heilai Huang \\& Xin Pei. Jointly modeling area-level crash rates by severity: A Bayesian multivariate random-parameters spatio-temporal Tobit regression. Transportmetrica A: Transport Science, 2019, 15(2): 1867-1884. https://doi.org/10.1080/23249935.2019.1652867.

[3] Zeng, Qiang, Wen Huiying, Wong S.C., Huang Helai, Guo Qiang, Pei Xin. Spatial joint analysis for zonal daytime and nighttime crash frequencies using a Bayesian bivariate conditional autoregressive model. Journal of Transportation Safety and Security, 2020, 12(4): 566-585. 10.1080/19439962.2018.1516259.

Besides, some more directions for future research are suggested to draw in the Conclusion Section.

Response. Thank your suggestion. We have added the above references in the revised version. The ideas of asymptotic and exact methods can be extended other data structures such as crash data. For the problem, it is worthy of researching and exploring (see the Conclusion Section).

Reviewer 2:

The topic of this paper is interesting. The methods sound. The results are meaningful and useful. There is one suggestion to improve this paper.

Some related references about likelihood ratio test or maximum likelihood estimations could be added.

[1] Dong Bowen, Ma Xiaoxiang, Chen Feng, Chen Suren. Investigating the Differences of Single- and Multi-vehicle Accident Probability Using Mixed Logit Model. Journal of Advanced Transportation, 2018, UNSP 2702360. DOI：10.1155/2018/2702360.

[2] Chen Feng, Chen Suren, Ma Xiaoxiang. Analysis of hourly crash likelihood using unbalanced panel data mixed logit model and real-time driving environmental big data. 2018, Journal of Safety Research. 65: 153-159. DOI：10.1016/j.jsr.2018.02.010.

[3] Feng Chen, Mingtao Song and Xiaoxiang Ma. Investigation on the Injury Severity of Drivers in Rear-End Collisions Between Cars Using a Random Parameters Bivariate Ordered Probit Model. International Journal of Environmental Research and Public Health, 2019, 16(14) , 2632. https://doi.org/10.3390/ijerph16142632.

[4] Chen Feng, Chen Suren, Ma Xiaoxiang. Crash Frequency Modeling Using Real-Time Environmental and Traffic Data and Unbalanced Panel Data Models. International Journal of Environmental Research and Public Health, 2016, 13(6), 609. DOI：10.3390/ijerph13060609.

Response. Thank your suggestion. Some related references have been added in our revision.

Yours Sincerely,

Zhiming Li

---

## [Decision Letter · Decision Letter 1]

9 Nov 2020

Statistical tests under Dallal's model: Asymptotic and exact methods

PONE-D-20-26283R1

Dear Dr. Li,

We’re pleased to inform you that your manuscript has been judged scientifically suitable for publication and will be formally accepted for publication once it meets all outstanding technical requirements.

Kind regards,

Feng Chen

Academic Editor

PLOS ONE

Additional Editor Comments (optional):

Reviewers' comments:

Reviewer's Responses to Questions

**Comments to the Author**

1. If the authors have adequately addressed your comments raised in a previous round of review and you feel that this manuscript is now acceptable for publication, you may indicate that here to bypass the “Comments to the Author” section, enter your conflict of interest statement in the “Confidential to Editor” section, and submit your "Accept" recommendation.

Reviewer #1: All comments have been addressed

2. Is the manuscript technically sound, and do the data support the conclusions?

Reviewer #1: (No Response)

3. Has the statistical analysis been performed appropriately and rigorously? 

Reviewer #1: (No Response)

4. Have the authors made all data underlying the findings in their manuscript fully available?

Reviewer #1: (No Response)

5. Is the manuscript presented in an intelligible fashion and written in standard English?

Reviewer #1: (No Response)

6. Review Comments to the Author

Reviewer #1: (No Response)

7. PLOS authors have the option to publish the peer review history of their article (what does this mean?). If published, this will include your full peer review and any attached files.

Reviewer #1: No

---

## [Editor Report · Acceptance letter]

16 Nov 2020

PONE-D-20-26283R1 

Statistical tests under Dallal’s model: Asymptotic and exact methods  

Dear Dr. Li:

I'm pleased to inform you that your manuscript has been deemed suitable for publication in PLOS ONE. Congratulations! Your manuscript is now with our production department. 

Kind regards, 

on behalf of

Dr. Feng Chen 

Academic Editor

PLOS ONE